# Benchmarking LHC searches for light 2HDM+$a$ pseudoscalars

**Spyros Argyropoulos**[1][⋆] **and Ulrich Haisch**[2][†]

**1** Physikalisches Institut, Universität Freiburg,
Hermann-Herder Str. 3a, 79104 Freiburg, Germany
**2** Werner-Heisenberg-Institut, Max-Planck-Institut für Physik,
Föhringer Ring 6, 80805 München, Germany

⋆ Spyridon.Argyropoulos@cern.ch † haisch@mpp.mpg.de

## Abstract

Using two suitable benchmark scenarios that satisfy the experimental constraints on the total decay width of the 125 GeV Higgs boson, we determine the bounds on light CP-odd spin-0 states in the 2HDM+$a$ model that arise from existing LHC searches. Our work represents the first thorough study that considers the parameter space with $m_a \lesssim 100$ GeV and should prove useful for 2HDM+$a$ interpretations of future ATLAS, CMS and LHCb searches for pseudoscalars with masses below the electroweak scale.

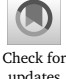

## 1 Motivation

The so-called 2HDM+$a$ model [1–4] is the simplest gauge-invariant and renormalisable extension of the simplified pseudoscalar dark matter (DM) model [5,6]. It includes a DM candidate in the form of a Dirac fermion $\chi$ that is a singlet under the Standard Model (SM) gauge group, four two-Higgs-doublet model (2HDM) spin-0 states and an additional CP-odd mediator that

provides the dominant portal between the dark and the visible sector. Since for models with pseudoscalar mediators the DM direct detection (DD) constraints are weaker compared to models with scalar mediators, such models are more attractive from an astrophysical point of view since they often allow to reproduce the observed DM relic abundance in a wider parameter space and with less tuning. These features admit a host of missing transverse momentum ($E_T^{\text{miss}}$) signatures at colliders which can be consistently compared and combined, making the 2HDM+$a$ model one of the pillars of the LHC DM search programme [7–23].

Besides $E_T^{\text{miss}}$ searches also direct searches for spin-0 states in SM final states can be used to explore and to constrain the 2HDM+$a$ parameter space. While the latter subject has received some attention [4,8,24–27] mostly focusing on heavy non-SM Higgses, the goal of this work is to study in detail the case of a light pseudoscalar $a$. In particular, we will discuss in Section 2 in which case searches for exotic decays of the 125 GeV Higgs of the form $h \to aa \to 4f$ [28–41] as well as measurements of dimuon cross sections targeting $pp \to a \to \mu^+ \mu^-$ [29, 42–50] provide valuable probes of the parameter space of the 2HDM+$a$ model at the LHC. Constraints from these channels have not been considered in the context of the 2HDM+$a$ model before. Our general findings will be illustrated in Section 3 by considering two suitable parameter benchmark scenarios as examples. For these two benchmark choices we derive the constraints from existing LHC searches and compare them to the regions in parameter space that allow to obtain the correct DM relic density assuming standard thermal freeze-out. The constraints from DM DD experiments are also discussed. We commence without further ado.

## 2 Theoretical framework in a nutshell

To make our work self-contained we begin this section with a comprehensive discussion of the structure of the 2HDM+$a$ model. Further details can be found in [4, 11, 21, 22]. In the 2HDM+$a$ the tree-level scalar potential can be written as

$$
\begin{aligned}
V_H = {}& \mu_1 H_1^\dagger H_1 + \mu_2 H_2^\dagger H_2 + \left( \mu_3 H_1^\dagger H_2 + \text{h.c.} \right) + \lambda_1 \left( H_1^\dagger H_1 \right)^2 + \lambda_2 \left( H_2^\dagger H_2 \right)^2 \\
& + \lambda_3 \left( H_1^\dagger H_1 \right) \left( H_2^\dagger H_2 \right) + \lambda_4 \left( H_1^\dagger H_2 \right) \left( H_2^\dagger H_1 \right) + \left[ \lambda_5 \left( H_1^\dagger H_2 \right)^2 + \text{h.c.} \right],
\end{aligned}
\tag{1}
$$

where we have imposed a $\mathbb{Z}_2$ symmetry under which the two Higgs doublets $H_1$ and $H_2$ transform as $H_1 \to H_1$ and $H_2 \to -H_2$ but allowed for this discrete symmetry to be broken softly by the terms $\mu_3 H_1^\dagger H_2 + \text{h.c.}$ The $\mathbb{Z}_2$ symmetry is the minimal condition necessary to guarantee the absence of flavour-changing neutral currents at tree level. To avoid possible issues with electric dipole moments, we furthermore assume that all parameters appearing in (1) are real. In such a case the CP eigenstates that emerge after spontaneous symmetry breaking from $V_H$ can be identified with the mass eigenstates: two CP-even scalars $h$ and $H$, one CP-odd pseudoscalar $A$ and one charged scalar $H^\pm$. Besides the masses of the five Higgs bosons, the 2HDM parameter space involves the angles $\alpha$ and $\beta$. The former angle describes the mixing of the two CP-even Higgs bosons while the latter encodes the ratio of the vacuum expectation values (VEVs) $v_1$ and $v_2$ of the two Higgs doublets $\tan \beta = v_2 / v_1$ with $v = (v_1^2 + v_2^2)^{1/2}$.

The tree-level scalar potential in the 2HDM+$a$ model contains besides (1) the following two contributions

$$
V_{HP} = P \left( i \, b_P H_1^\dagger H_2 + \text{h.c.} \right) + P^2 \left( \lambda_{P1} H_1^\dagger H_1 + \lambda_{P2} H_2^\dagger H_2 \right), \qquad V_P = \frac{1}{2} m_P^2 P^2 .
\tag{2}
$$

The parameters $b_P$, $\lambda_{P1}$ and $\lambda_{P2}$ are taken to be real to not violate CP. Notice that the first term in $V_{HP}$ breaks the $\mathbb{Z}_2$ symmetry softly. A quartic term $P^4$ is not included in $V_P$ since it does not

lead to any substantial modification of the LHC phenomenology, in particular such an addition would have no impact on all observables discussed in this work.

If the Dirac DM field $\chi$ and the pseudosalar $P$ are taken to transform under the $\mathbb{Z}_2$ symmetry as $\chi \rightarrow -\chi$ and $P \rightarrow P$, the only renormalisable DM-mediator coupling that is allowed by symmetry is

$$\mathcal{L}_\chi = -i y_\chi P \bar{\chi} \gamma_5 \chi \,, \tag{3}$$

where it is assumed again that the dark-sector Yukawa coupling $y_\chi$ is real, making all non-SM interactions in the 2HDM+$a$ model CP conserving.

Including the mass $m_\chi$ of the DM particle, the Lagrangian of the 2HDM+$a$ model contains 14 free parameters in addition to the SM ones. After rotation to the mass eigenbasis, these 14 parameters can be traded for seven physical masses, three mixing angles and four couplings:

$$\begin{Bmatrix} \mu_1, \mu_2, \mu_3, b_P, m_P, m_\chi, \\ y_\chi, \lambda_1, \lambda_2, \lambda_3, \lambda_4, \lambda_5, \\ \lambda_{P1}, \lambda_{P2} \end{Bmatrix} \iff \begin{Bmatrix} v, m_h, m_A, m_H, m_{H^\pm}, m_a, m_\chi, \\ \cos(\beta - \alpha), \tan\beta, \sin\theta, \\ y_\chi, \lambda_3, \lambda_{P1}, \lambda_{P2} \end{Bmatrix} \,. \tag{4}$$

Here $\sin\theta$ represents the mixing of the two CP-odd weak spin-0 eigenstates and the additional CP-even mediator $a$ is mostly composed of $P$ for $\sin\theta \simeq 0$. The parameters appearing on the right-hand side of (4) are used as input in the analyses of the 2HDM+$a$ model. Since the VEV $v \simeq 246\,\text{GeV}$ and the Higgs mass $m_h \simeq 125\,\text{GeV}$ are already fixed by observations there are 12 input parameters.

As in [7–23] we choose the Yukawa sector of the 2HDM+$a$ model to be of type-II. In the so-called alignment limit, i.e. assuming $\cos(\beta - \alpha) = 0$, which guarantees that the Higgs $h$ is SM-like, the neutral non-SM scalars couple to the SM fermions in the following way

$$g_{Hf\bar{f}} \propto y_f \eta_f \,, \qquad g_{Af\bar{f}} \propto y_f \eta_f \cos\theta \,, \qquad g_{af\bar{f}} \propto y_f \eta_f \sin\theta \,. \tag{5}$$

Here $y_f = \sqrt{2} m_f / v$ denotes the SM Yukawa couplings and $\eta_u = \cot\beta$ and $\eta_d = \eta_\ell = \tan\beta$ in the case of up-, down-type quarks and charged leptons, respectively.

If the spin-0 mediator $a$ that provides the dominant link between the dark and the visible sector in the 2HDM+$a$ model is sufficiently light, the 125 GeV Higgs boson $h$ discovered at the LHC can decay into a pair of such CP-odd states. The corresponding partial decay width can be written as

$$\Gamma(h \rightarrow aa) = \frac{g_{haa}^2 m_h}{32\pi} \sqrt{1 - \frac{4m_a^2}{m_h^2}} \,, \tag{6}$$

where $m_a$ denotes the relevant pseudoscalar mass. In the alignment limit and assuming degenerate 2HDM heavy Higgs masses, i.e. $m_A = m_H = m_{H^\pm}$, the coupling $g_{haa}$ in (6) takes the form [4]

$$g_{haa} = \frac{1}{m_h v} \left[ 2 \left( m_A^2 - m_a^2 + \frac{m_h^2}{2} - \lambda_3 v^2 \right) \sin^2\theta \right.$$
$$\left. - 2 \left( \lambda_{P1} \cos^2\beta + \lambda_{P2} \sin^2\beta \right) v^2 \cos^2\theta \right]. \tag{7}$$

If $m_\chi < m_a/2$ with $m_\chi$ the mass of the fermionic DM candidate in the 2HDM+$a$ model, the decay $h \rightarrow aa$ followed by $a \rightarrow \chi\bar{\chi}$ will be lead to an invisible Higgs decay signal ($h \rightarrow \text{inv}$). The latest searches for invisible decays of the Higgs boson [51, 52] impose a lower limit of $m_a \gtrsim 100\,\text{GeV}$ on the pseudoscalar mass [4] unless the coupling $g_{haa}$ is tuned so that

$|g_{haa}| \lesssim 0.02.$[1] On the other hand, for $m_\chi > m_a/2$ the decay channel $a \to \chi\bar{\chi}$ is closed and $h \to \text{inv}$ provides no constraints. Exotic Higgs decays are however still possible in such a case since the pseudoscalar can decay via $a \to f\bar{f}$ to all kinematically accessible SM fermions, i.e. those with $m_f < m_a/2$. This opens up the possibility to constrain 2HDM+$a$ realisations with a light pseudoscalar $a$ through direct measurements of $\Gamma_h$ [22]. In fact, using (6) and recalling that the total Higgs decay width in the SM is with $\Gamma_h^{\text{SM}} \simeq 4.07\,\text{MeV}$ [53] much smaller than the LHC sensitivity on $\Gamma_h$, one can derive the following approximate inequality

$$|g_{haa}| \lesssim \sqrt{\frac{32\pi\Gamma_h}{m_h}}. \tag{8}$$

For the best 95% confidence level (CL) bound of $\Gamma_h < 1.1\,\text{GeV}$ that derives at present from the direct measurements of the total Higgs width [54,55], the result (8) implies

$$|g_{haa}| \lesssim 0.94. \tag{9}$$

We add that indirect determinations of the total Higgs decay width at the LHC using off-shell Higgs production [56–61] lead nominally to bounds on $\Gamma_h$ that are significantly more stringent than those that derive from the direct measurements [54,55]. While it is tempting to use the indirect determinations of $\Gamma_h$ to set bounds on $g_{haa}$ a la (9) it is important to realise that the limits on $\Gamma_h$ derived in [56–61] are model dependent since they all assume a certain connection between the on- and off-shell Higgs production rates. This connection is established by noting that rescaling the SM Higgs couplings $g_{hXX}^{\text{SM}}$ and $\Gamma_h^{\text{SM}}$ as follows $g_{hXX}^{\text{SM}} \to \xi^{1/4}\,g_{hXX}^{\text{SM}}$ and $\Gamma_h^{\text{SM}} \to \xi\,\Gamma_h^{\text{SM}}$ leaves the on-shell Higgs production rate unchanged but modifies the kinematic distributions in $pp \to h^* \to ZZ \to 4\ell$ production. The stringent limits on $\Gamma_h$ obtained in [56–61] only apply in models in which the modifications of $g_{hXX}$ and $\Gamma_h$ are at least approximately compatible with the above rescaling relations. Clearly, this is not the case in the 2HDM+$a$ model where the couplings $g_{hXX}$ are unmodified in the alignment limit, but $\Gamma_h$ is altered because of the presence of additional decay channels such as $h \to aa$ with $a \to \chi\bar{\chi}, f\bar{f}$.

## 3 Numerical study and discussion

From the earlier discussion it follows that in order for the processes $h \to aa \to 4f$ and $pp \to a \to \mu^+\mu^-$ to provide relevant constraints on the 2HDM+$a$ parameter space one has to dial the 2HDM+$a$ parameters entering (7) such that the coupling $g_{haa}$ fulfils the bound (8) or equivalent (9). While this requires always some tuning, suitable benchmark scenarios can be simply obtained from the recommendations in the LHC DM Working Group (LHCDMWG) white paper [11]. The parameter choices $\cos(\beta - \alpha) = 0$, $m_A = m_H = m_{H^\pm}$ and $\lambda_3 = \lambda_{P1} = \lambda_{P2}$ are common to the benchmarks studied in the following and we furthermore employ a Yukawa sector of type-II throughout this work. As a result the couplings of the pseudoscalar $a$ to up-, down-type quarks and charged leptons behave as $g_{au\bar{u}} \propto \cot\beta$ and $g_{ad\bar{d}} \propto \tan\beta$ and $g_{a\ell^+\ell^-} \propto \tan\beta$, respectively (see (5)). Constraints are then derived in the $m_a - m_\chi$ plane keeping the parameters $m_A$, $\tan\beta$, $\sin\theta$, $\lambda_3$ and $y_\chi$ fixed in each scan.

### 3.1 Benchmark I

The first 2HDM+$a$ benchmark scenario that we will study as an example to illustrate the typical constraints that derive from the $h \to aa \to 4f$ and $pp \to a \to \mu^+\mu^-$ processes is:

$$\{m_A, \tan\beta, \sin\theta, \lambda_3, y_\chi\} = \{1.2\,\text{TeV}, 1, 0.35, 3, 1\}, \quad (\text{benchmark I}). \tag{10}$$

---

[1]Notice that this bound is stronger than $m_a > m_h/2$, that one would naively expect, due to the off-shell contributions to the four-body decays of the form $h \to aa \to \chi\bar{\chi}f\bar{f}$ with $f$ denoting a SM fermion. Given the very small total SM Higgs decay width $\Gamma_h^{\text{SM}}$ these contributions are phenomenologically relevant [4].

Notice that the value of $m_A$ has been chosen such that the benchmark leads to a value of $\Gamma(h \to aa)$ consistent with the current bound (9) on $|g_{haa}|$ assuming a light pseudoscalar $a$. This corresponds to a parameter tuning of around 10% because in the case of (10) the inequality (9) is only satisfied for $m_A \in [1160, 1270]\,$GeV. Since the benchmark I represents a slight variation of one of the standard parameter choices recommended by the LHCDMWG in [11] all constraints that arise from Higgs and flavour physics, electroweak precision measurements and vacuum stability are automatically fulfilled for the choices (10). See the works [4, 11, 21, 22] for details.

In the right panel of Figure 1 we show an assortment of constraints in the $m_a$–$m_\chi$ plane for the 2HDM+$a$ parameters (10). One observes from the yellow contours that the existing $h \to aa \to 4f$ searches exclude almost the entire parameter space with $m_a \in [1, 62]\,$GeV and $m_\chi > m_a/2$ at 95% CL.[2] Only small mass windows close to the $J/\psi$ and $\Upsilon$ resonances remain allowed since these mass ranges are vetoed in all experimental analyses. For $m_a < 10\,$GeV the most constraining searches are $h \to aa \to 4\mu$ [33, 39, 40], $h \to aa \to 2\mu 2\tau$ [37] and $h \to aa \to 4\tau$ [28, 35], while in the case of $m_a > 10\,$GeV the searches for $h \to aa \to 2\mu 2b$ [38], $h \to aa \to 2\tau 2b$ [31] and $h \to aa \to 4b$ [32] provide the leading bounds at present. The relevance of these decay modes can be understood by looking at the left panel in Figure 1 which displays the branching ratios of the pseudoscalar $a$ as a function of its mass for decoupled DM, i.e. assuming $m_\chi > m_a/2$. We emphasise that the shown results correspond to a leading-order perturbative calculation. In particular, non-perturbative effects that are relevant for $m_a \lesssim 3\,$GeV as well as in the vicinity of $m_a \simeq m_{J/\psi}$ and $m_a \simeq m_\Upsilon$ are not included. For details see for instance [29, 43, 68]. This simplification has no impact on the constraints displayed on the right-hand side. The search for narrow resonances in $pp \to a \to \mu^+\mu^-$ production by LHCb [50] furthermore excludes the parameter space with $m_a \in [1, 8]\,$GeV and $m_\chi > m_a/2$. The corresponding 90% CL is displayed as a red vertical line. The production cross sections needed to extract this limit have been calculated at leading order in QCD with the help of `MadGraph5_aMCNLO` [69] using `NNPDF31_nlo_as_118` parton distribution functions [70]. The constraints due to the BaBar searches for dimuon pairs in radiative decays of $\Upsilon$ mesons [62, 63] are shaded purple and lead to the 90% CL limit $m_a \in [1, 2.5]\,$GeV for $m_\chi > m_a/2$. Our recast relies in this case on the methodology described in Appendix A of [71]. We add that light pseudoscalars $a$ with $m_a \lesssim 10\,$GeV are also subject to the constraints of various other rare $B$- and $K$-meson decays (see for example [71]). For better readability these bounds have not been included in our figure.

In order to probe the parameter region with $m_\chi < m_a/2$ we consider the latest searches for $h \to \text{inv}$ [51, 52] that imply $\text{Br}(h \to \text{inv}) < 0.11$ and the $h + E_T^{\text{miss}}$ analysis in the $h \to b\bar{b}$ channel $\left(h(b\bar{b}) + E_T^{\text{miss}}\right)$ [18]. The corresponding 95% CL exclusions are shown in blue and green in the right panel of Figure 1, respectively. The expected sensitivity of the $h(b\bar{b}) + E_T^{\text{miss}}$ signal is estimated from the model-independent upper limits on the visible cross section and the product $\mathcal{A} \cdot \epsilon$ of the signal acceptance $\mathcal{A}$ and reconstruction efficiency $\epsilon$ provided by ATLAS in the auxiliary material of [18]. In this way upper limits on the signal strength in each of the analysis regions are derived that are then statistically combined to obtain the total expected sensitivity. In this combination it is assumed that the signal contributions in the different analysis regions are independent of each other. We add that the $h(b\bar{b}) + E_T^{\text{miss}}$ search [18] provides at present the most stringent mono-$X$ constraint for the 2HDM+$a$ parameter scenario (10). This can be interfered for instance from the left plot in Figure 7 of the review article [22]. Notice that the $h \to \text{inv}$ and the $h(b\bar{b}) + E_T^{\text{miss}}$ search are complementary to each other in the

---

[2]The parameter space with $m_a \lesssim m_h/2 \simeq 62.5\,$GeV is also disfavoured by the measurements of the global signal strength $\mu$ of the 125 GeV Higgs boson [66, 67]. The bounds on the individual branching ratios from the $h \to aa \to 4f$ searches are however more stringent and direct than the rather indirect limit arising from the determinations of $\mu$.

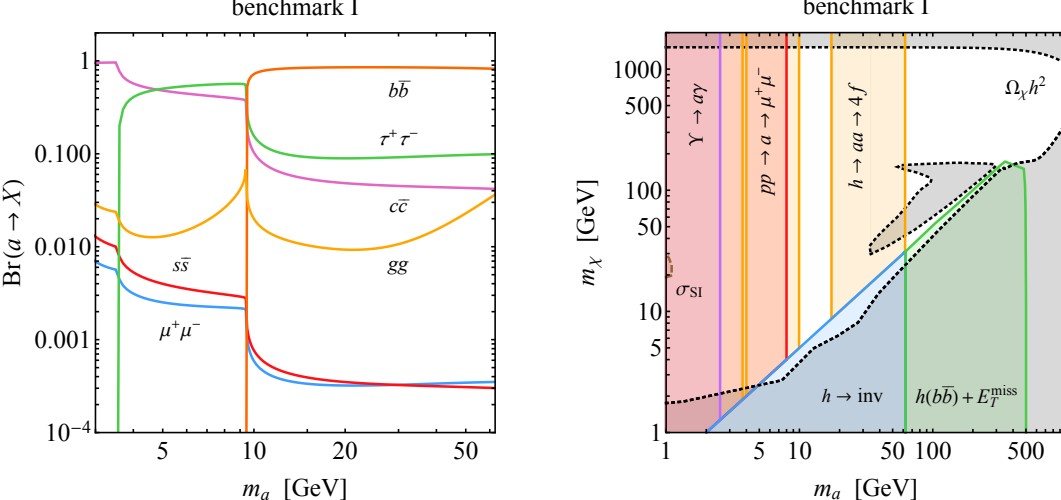

Figure 1: Left: Branching ratios of the pseudoscalar $a$ as a function of its mass in the benchmark I model assuming that the decays $a \to \chi \bar{\chi}$ is kinematically forbidden. Right: Constraints in the $m_a$–$m_\chi$ plane for benchmark I in the 2HDM+$a$ model. The yellow contours represent the combined bound from the existing $h \to aa \to 4f$ searches [28, 30–40], while the red contour corresponds to the best limits due to the available dimuon cross section measurements at the LHC [42, 45, 47–50]. The parameter region shaded purple is excluded by the dimuon searches in radiative $\Upsilon$-mesons decays [62,63]. The blue contour is the exclusion that derives from the latest searches for $h \to$ inv [51, 52], while the constraint that derives from a recast of the $h + E_T^{\mathrm{miss}}$ analysis in the $h \to b\bar{b}$ channel [18] is shown in green. The black dotted contours indicate the parameter choices for which the correct DM relic density $\Omega_\chi h^2 = 0.12$ [64] is achieved. In the gray shaded areas DM is overabundent. The measurement of the spin-independent (SI) DM-nucleon cross section $\sigma_{\mathrm{SI}}$ [65] disfavours the shaded region inside the brown dashed curve. See main text for further details.

sense that only the region with $m_a < m_h/2 \simeq 62.5\,\mathrm{GeV}$ and $m_\chi < m_a/2$ is kinematically accessible to the former, while the latter tests the parameter space with $m_a > m_h/2 \simeq 62.5\,\mathrm{GeV}$ and $m_\chi < m_a/2$, because $\mathrm{Br}\left(h \to b\bar{b}\right)$ becomes very small for $m_a < m_h/2$.

The parameter sets in the $m_a$–$m_\chi$ plane for which the DM relic density $\Omega_\chi h^2 = 0.12$ as measured by Planck [64] is obtained are indicated by the black dotted contours in the right panel of Figure 1. Areas with DM overproduction are shaded gray. The DM relic density calculation is performed using MadDM [72] and relies on the simplified assumption that $\Omega_\chi h^2$ is solely determined by the interactions predicted in the 2HDM+$a$ model. The observed DM relic abundance can be achieved in three distinct regions of parameter space. For $m_\chi \simeq m_a/2$ where DM annihilation via $\chi \bar{\chi} \to a \to f\bar{f}$ is resonantly enhanced as well as in the island just above the diagonal $m_\chi = m_a/2$ where $\chi \bar{\chi} \to ha$ followed by SM Higgs decays and $a \to f\bar{f}$ sets $\Omega_\chi h^2$. Notice that in the former case $\chi \bar{\chi} \to b\bar{b}$ dominates for $m_a \lesssim 350\,\mathrm{GeV}$, while for $m_a \gtrsim 350\,\mathrm{GeV}$ annihilation via $\chi \bar{\chi} \to t\bar{t}$ also plays an important role and leads to a rise of the relic density contour. Finally, at $m_\chi \simeq 1.5\,\mathrm{TeV}$ it is again possible to obtain $\Omega_\chi h^2 = 0.12$ for low pseudoscalar masses $m_a$. In this case the dominant annihilation channels are $\chi \bar{\chi} \to hA$, $\chi \bar{\chi} \to ZH$ and $\chi \bar{\chi} \to W^\pm H^\mp$ with the final-state bosons subsequently decaying to SM fermions.

Although loop suppressed in the 2HDM+$a$ model [10, 73–75] we also consider the constraints that DM DD experiments like XENON1T [65] set in the benchmark I scenario. The ob-

ject of interest in this case is the SI DM-nucleon cross section that can be approximated by

$$\sigma_{\text{SI}} \simeq \left( \frac{m_N \, m_\chi}{m_N + m_\chi} \right)^2 \frac{c_N^2}{\pi}, \tag{11}$$

with $m_N \simeq 939 \, \text{MeV}$ the average of the nucleon mass and $c_N$ the Wilson coefficient of the dimension-six nucleon operator $O_N = \chi \bar{\chi} \bar{N} N$. As explained in detail in [74, 75] the Wilson coefficient $c_N$ receives in general contributions from Higgs-induced one-loop triangle and box diagrams as well as two-loop contributions leading to effective DM-gluon interactions. By utilising the results of the calculation [75] we find that for benchmark I the effects of one-loop box diagrams and two-loop graphs with bottom- and charm-quark loops are below a percent in the relevant parameter space. Neglecting these contributions and evaluating the two-loop top-quark corrections in the infinite mass limit, it then turns out that the Wilson coefficient $c_N$ can be very well approximated by the following simple expression:

$$c_N \simeq \frac{y_\chi^2 \sin^2 \theta}{16 \pi^2} \frac{m_N}{m_\chi} \left( \frac{2 g_{ahh} f_N}{m_h^2} + \frac{f_G}{v^2} \cos^2 \theta \right) C^{(1)}(x_{a/\chi}). \tag{12}$$

Here $g_{haa}$ has already been given in (7) and we have introduced $x_{i/j} = m_i^2 / m_j^2$. In the case of the 2HDM+$a$ model of type-II the effective interaction strengths $f_N$ and $f_G$ are given by

$$f_N = \sum_{q=u,d,s} f_{T_q}^N + \frac{2}{9} f_{T_G}^N, \qquad f_G = \frac{2 \cot^2 \beta}{27} f_{T_G}^N, \tag{13}$$

where $f_{T_u}^N \simeq 0.019$, $f_{T_d}^N \simeq 0.045$ and $f_{T_s}^N \simeq 0.043$ [76–79] are the quark-nucleon matrix elements and $f_{T_G}^N = 1 - \sum_{q=u,d,s} f_{T_q}^N \simeq 0.89$ is the effective gluon-nucleon coupling. The one-loop triangle form factor finally reads

$$C^{(1)}(x) = \frac{(3-x)\sqrt{x} \, \ln\left( \frac{1}{2} \left[ \sqrt{x-4} + \sqrt{x} \right] \right)}{\sqrt{x-4}} + \frac{1}{2}(x-1)\ln x - 1$$
$$\simeq \begin{cases} -\frac{\ln x + 2}{2}, & x \to 0, \\[2mm] \frac{1}{2x}, & x \to \infty. \end{cases} \tag{14}$$

The above formulae can be used to translate the latest XENON1T 90% CL upper limit on the SI DM-nucleon cross section [65] into constraints on the 2HDM+$a$ parameter space. The tiny shaded region at $m_a \simeq 1 \, \text{GeV}$ and $m_\chi \simeq 25 \, \text{GeV}$ in the right panel of Figure 1 that is enclosed by a brown dashed curve corresponds to the exclusion found in the case of the benchmark I model. Notice that the DM DD constraint is so weak not only because it is suppressed by a loop factor and $\sin^2 \theta$ but also because the coupling $g_{ahh}$ that enters (12) fulfils the bound (9). In fact, in the case at hand the contributions proportional to $f_N$ and $f_G$ in (12) interfere destructively, since $g_{haa} < 0$ in the relevant parameter region, which leads to a further suppression. While DM DD experiments hence do not provide meaningful constraints on (10) we have decided to keep the formulae (11) to (14) because they allow for a straightforward evaluation of $\sigma_{\text{SI}}$ for all 2HDM+$a$ model realisation of type-II with $m_A \gg m_a$ and sufficiently small values of $\tan \beta$. They hence can be applied in the majority of the benchmark scenarios recommended in the LHCDMWG white paper [11].

## 3.2 Benchmark II

To further demonstrate the constraining power of $h \to aa \to 4f$ and $pp \to a \to \mu^+ \mu^-$ searches in 2HDM+$a$ model realisations that satisfy the upper bound (9), we consider the following

parameter choices:

$$\{m_A, \tan\beta, \sin\theta, \lambda_3, y_\chi\} = \{1.0\,\text{TeV}, 40, 0.7, 8, 0.1\}, \quad (\text{benchmark II}). \qquad (15)$$

Notice that the parameters chosen in benchmark II give rise to one of the parameter scenarios that has been recently studied in [80] and aims to explain a possible excess in the measurement of the anomalous magnetic moment $a_\mu = (g-2)_\mu/2$ of the muon. We add that for the parameter choice in (15) to satisfy (9), the heavy Higgs mass $m_A$ has to lie within the range $m_A \in [975, 1005]\,\text{GeV}$, which corresponds to a tuning of parameters of around 3%. While constraints from flavour physics [80], electroweak precision measurements and vacuum stability are again fulfilled for benchmark II, the width of the heavy CP-even Higgs turns out to be very large since the decay $H \to aa$ is unsuppressed and kinematically allowed for $m_a < 500\,\text{GeV}$ with $g_{Haa}$ becoming non-perturbative in a large region of parameter space. Since $a \to b\bar{b}$ is the dominant decay mode of the pseudoscalar this may result in an observable $4b$ signature that we however do not attempt to calculate because of the large value of $\Gamma_H$.

Since this article mainly addresses the LHC physics practitioner let us briefly explain how a light pseudoscalar $a$ contributes to $a_\mu$ in the 2HDM+$a$ model of type-II. See also also [80–87] for related discussions. The virtual exchange of a pseudoscalar $a$ leads to a correction to $a_\mu$ at the one-loop level. The corresponding contribution is given by [81]

$$\delta a_\mu^{(1)} = -\frac{\alpha}{8\pi \sin^2\theta_w} \frac{m_\mu^4}{m_W^2 m_a^2} \sin^2\theta \tan^2\beta \, F^{(1)}\big(x_{\mu/a}\big), \qquad (16)$$

where $\alpha \simeq 1/137$ is the electromagnetic fine-structure constant, $m_W \simeq 80.4\,\text{GeV}$ denotes the $W$-boson mass and $\sin^2\theta_w \simeq 0.23$ is the sine squared of the weak mixing angle. The one-loop form factor appearing in (16) takes the form

$$F^{(1)}(x) = \int_0^1 dz \, \frac{z^3}{1-z+z^2 x} \simeq \begin{cases} -\ln x - \frac{11}{6}, & x \to 0, \\[2mm] \frac{1}{2x}, & x \to \infty. \end{cases} \qquad (17)$$

Since the one-loop correction (16) is strongly Yukawa suppressed by a factor of $m_\mu^4$, two-loop diagrams of Barr-Zee type [88] can be numerically important and even larger than the one-loop contribution. The dominant two-loop correction of Barr-Zee type involves the exchange of a pseudoscalar $a$ and a photon and takes the following form [81][3]

$$\delta a_\mu^{(2)} = \frac{\alpha^2}{8\pi^2 \sin^2\theta_w} \frac{m_\mu^2}{m_W^2} \sin^2\theta \sum_f c_f \frac{m_f^2}{m_a^2} F^{(2)}\big(x_{f/a}\big), \qquad (18)$$

in the 2HDM+$a$ model. Notice that (18) contains contributions that are parametrically enhanced with respect to (16) by a factor of $m_f^2/m_\mu^2$. Barr-Zee type diagrams with internal $Z$-boson exchange also exist but their contribution is numerically insignificant because they are suppressed by the vector coupling of the $Z$ boson to muons $1 - 4\sin^2\theta_w \simeq 0.08$. In (18) the sum over $f$ includes all SM fermions and we have introduced the coefficients $c_u = 4/3$, $c_d = \tan^2\beta/3$ and $c_\ell = \tan^2\beta$ for up-, down-type quarks and charged leptons, respectively. The relevant two-loop form factor reads

$$F^{(2)}(x) = \int_0^1 dz \, \frac{\ln\big(\frac{x}{z(1-z)}\big)}{x-z(1-z)} \simeq \begin{cases} \ln^2 x + \frac{\pi^2}{3}, & x \to 0, \\[2mm] \frac{\ln x + 2}{x}, & x \to \infty. \end{cases} \qquad (19)$$

---

[3]The formulae in [80] that correspond to our results for $\delta a_\mu^{(2)}$ and $F^{(2)}(x)$ contain two typographical mistakes.

Employing now the benchmark II parameter choices (15) together with $m_a = 10\,\text{GeV}$ one finds from (16) to (19) that $\delta a_\mu^{(1)} \simeq -1.4 \cdot 10^{-9}$ and $\delta a_\mu^{(2)} \simeq 3.8 \cdot 10^{-9}$. The total 2HDM+$a$ contribution to the anomalous magnetic moment of the muon is thus $\delta a_\mu = \delta a_\mu^{(1)} + \delta a_\mu^{(2)} \simeq 2.4 \cdot 10^{-9}$. For the chosen parameters the 2HDM+$a$ corrections to $a_\mu$ therefore just have the right sign and size to explain the $4.2\sigma$ discrepancy between experiment [89, 90] and the SM prediction endorsed by the muon $g - 2$ theory initiative [91]:

$$\delta a_\mu = a_\mu^{\text{exp}} - a_\mu^{\text{SM}} = (2.51 \pm 0.59) \cdot 10^{-9} \,. \tag{20}$$

We add that the BMW collaboration has presented a new lattice-QCD evaluation of the hadronic vacuum polarisation contribution to $a_\mu$ [92]. If the BMW value of the hadronic vacuum polarisation is used to predict a SM the deviation in (20) is reduced to $1.6\sigma$, meaning that there is no particular evidence for a discrepancy with experiment. Notice finally that in order to enhance $a_\mu$ in the 2HDM+$a$ model of type-II the positive two-loop contribution (18) has to outweigh the negative one-loop correction (16). This generically only happens for parameter choices with $\tan\beta = \mathcal{O}(50)$ and $m_a = \mathcal{O}(10\,\text{GeV})$.

In the left and right panel of Figure 2 we show for benchmark II the branching ratios of the pseudoscalar $a$ as a function of $m_a$ assuming that $m_\chi > m_a/2$ and the most relevant constraints in the $m_a - m_\chi$ plane, respectively. From the right panel one observes that the ranges in $m_a$ that are excluded by the combination of the $h \to aa \to 4f$ searches [28, 30–40] in benchmark II resemble those that are also disfavoured in the case of benchmark I. However, in the case of benchmark II the exclusions that are set by $h \to aa \to 4f$ do not stop at $m_\chi = m_a/2$ but extend down to low DM masses. This feature is readily understood by noticing that for the benchmark II parameter choices (15) the invisible decay width of the pseudoscalar $a$ is strongly suppressed, i.e. $\Gamma(a \to \chi\bar{\chi}) \propto y_\chi^2 \cos^2\theta \simeq 0.005$, meaning that in benchmark II even pseudoscalars with $m_a > 2m_\chi$ have large branching ratios into SM fermions. This feature also explains why in the case of (15) the $h \to \text{inv}$ bound [51, 52] covers only the small triangular region with $m_a \in [0, 3.5]\,\text{GeV}$ and $m_\chi < m_a/2$, and why the $h(b\bar{b}) + E_T^{\text{miss}}$ search [18] leads to no relevant constraint.[4] One also sees that the searches for $pp \to a \to \mu^+\mu^-$ production [42, 45, 47–50] allow to put severe constraints on the parameter space of benchmark II. The constraints from the dimuon searches are so powerful in this case because the production cross sections $gg \to a$ and $b\bar{b} \to a$ are again enhanced by a factor $\sin^2\theta \tan^2\beta \simeq 780$. The same enhancement factor also appears in the partial decay rate $\Gamma(a \to \mu^+\mu^-)$. At present the most relevant $pp \to a \to \mu^+\mu^-$ searches are [49] and [50] which provide the leading constraints for $m_a \gtrsim 20\,\text{GeV}$ and $m_a \lesssim 8\,\text{GeV}$, respectively. In the mass region $m_a \in [11.5, 20]\,\text{GeV}$ both searches have similar sensitivities. We add that the gluon-gluon fusion channel represents the dominant production process in benchmark II for $m_a \lesssim 25\,\text{GeV}$, while for $m_a \gtrsim 25\,\text{GeV}$ bottom-quark fusion is the main production mechanism. The BaBar constraint from radiative $\Upsilon$ decays [62, 63] is also stronger in benchmark II than in benchmark I.

From the right panel in Figure 2 it is also evident that a combination of the limits stemming from the considered LHC searches almost entirely rules out the parameter space with $m_a \in [3.4, 20.1]\,\text{GeV}$ that leads to an explanation of the excess (20) of the measured value of $a_\mu$ compared its SM prediction. The relevant $m_a$ range is indicated by a yellow dash-dotted vertical band in the figure. The only viable mass ranges are presently $m_a \in [9.2, 9.7]\,\text{GeV}$, $m_a \in [9.8, 10.1]\,\text{GeV}$ and $m_a \in [10.2, 10.5]\,\text{GeV}$, but improved searches for a dimuon resonance in the $\Upsilon$ mass region [43, 47] might further reduce the allowed parameter space or even fully close it. Let us add in this context that benchmark II is in principle also excluded

---

[4]We also note that in the case of the $h(b\bar{b}) + E_T^{\text{miss}}$ signature, benchmark II leads to a much softer $E_T^{\text{miss}}$ spectrum than benchmark I, which also affects the sensitivity.

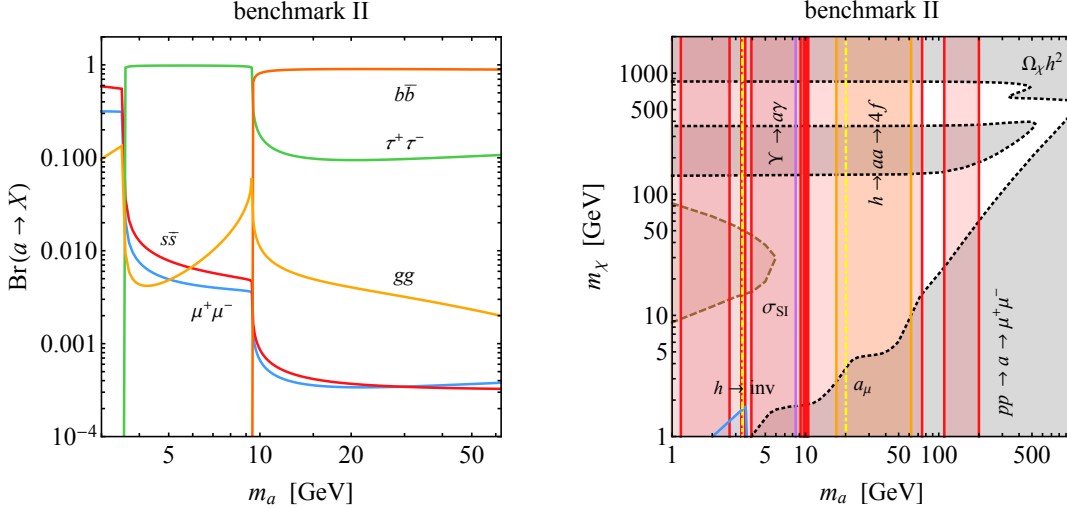

Figure 2: As Figure 1 but for the 2HDM+$a$ benchmark II scenario (15). Parameter choices inside the yellow dash-dotted vertical band shown in the right panel allow to accommodated the excess (20) observed in $a_\mu$ at the 95% CL. For additional explanations consult main text.

by the search $pp \to A \to \tau^+\tau^-$ [93] for all the values of $m_a$ that are shown in the right panel of Figure 2. It is therefore questionable if one can find a 2HDM+$a$ type-II model realisation that can explain the $a_\mu$ anomaly and survive the existing LHC constraints from non-SM Higgs production. Notice that in the case of a lepto-specific 2HDM+$a$ model of type-X the latter conclusion does not hold and addressing (20) is possible in the 2HDM+$a$ context [80].

The black dotted curves in the right panel of Figure 2 correspond to the contours with $\Omega_\chi h^2 = 0.12$ as measured by Planck [64]. As in benchmark I the observed DM relic abundance can be achieved in three separate regions in the $m_a$–$m_\chi$ plane. For not too heavy DM with $m_\chi \lesssim 500\,\text{GeV}$ annihilation proceeds dominantly via $\chi\bar\chi \to a \to b\bar b$. This process therefore sets $\Omega_\chi h^2$ in the bulk region. In the island at $m_\chi \simeq 250\,\text{GeV}$ also $\chi\bar\chi \to ha$ can provide a relevant source of wash-out in particular for light pseudoscalars $a$. For large DM masses multiple channels contribute to DM annihilation with $\chi\bar\chi \to ha$ and $\chi\bar\chi \to hA$ being the most important reactions. As illustrated by the brown dashed curve in the right panel in Figure 2, compared to benchmark I the constraints that arise from XENON1T [65] are stronger in benchmark II. The dominant correction to $\sigma_{\text{SI}}$ arise in this case from Higgs-induced one-loop triangle diagrams and two-loop contributions involving bottom quarks. While the former contribution can be calculated by considering the term proportional to $f_N$ in (12) to correctly include the two-loop bottom-quark contribution one has to perform the full calculation [74, 75]. While estimating the two-loop bottom-quark contribution by using (12) together with (13) but replacing $\cot^2\beta$ by $\tan^2\beta$ in $f_G$ is not a good approximation, such a replacement allows one to understand qualitatively why the DM direct detection constraints are more stringent in benchmark II than in benchmark I.

## 3.3 Final words

The main conclusion that can be drawn from the numerical results presented in this work is that LHC searches for $h \to aa \to 4f$ and $pp \to a \to \mu^+\mu^-$ production can provide interesting and complementary constraints on 2HDM+$a$ model realisations that feature a light pseudoscalar $a$. In particular, we have shown that the latter two types of processes can lead to relevant

constraints even in scenarios with a light pseudoscalar $a$ in which the stringent limits from $\text{Br}(h \to \text{inv})$ and $\Gamma_h$ are evaded by tuning the coupling $g_{haa}$ such that (8) or equivalently (9) is satisfied. To emphasise this generic finding we have studied two distinct parameter benchmarks and explored the sensitivity of the most relevant collider searches to them by performing parameter scans in the $m_a - m_\chi$ plane. The results of these scans are displayed in the right panels of Figures 1 and 2. To make contact to the astrophysical constraints on DM, we have also indicated in these two-dimensional scans the bounds that arise from DD experiments and the requirement to obtain the measured relic abundance. One important feature that is nicely illustrated in our scans is that LHC searches for $h \to aa \to 4f$ and $pp \to a \to \mu^+\mu^-$ production can probe regions of parameter space that lead to the correct value of $\Omega_\chi h^2$ but lie in the off-shell region $m_\chi > m_a/2$ and are therefore not accessible with mono-$X$ searches. In the context of the 2HDM+$a$ model of type-II and large $\tan\beta$ we have also argued that an explanation of a possible excess in the measurement of the anomalous magnetic moment $a_\mu$ of the muon (20) is generically at odds with the bounds from $h \to aa \to 4f$ and $pp \to a \to \mu^+\mu^-$ and possibly other non-SM Higgs search results. We believe that the results presented in our work should prove useful for 2HDM+$a$ interpretations of future ATLAS, CMS and LHCb searches for pseudoscalars $a$ with masses below the electroweak scale. In particular, by combining the direct measurements of the total Higgs decay width $\Gamma_h$ with the searches for $h \to \text{inv}$, $h \to aa \to 4f$ and $pp \to a \to \mu^+\mu^-$ it should be possible to exclude large parts of the 2HDM+$a$ model parameter space with $m_a \lesssim m_h/2 \simeq 62.5\,\text{GeV}$ in a model-independent fashion.

## Acknowledgements

We thank Fatih Ertas and Felix Kahlhoefer for providing us with a `Mathematica` implementation of the results presented in the publication [75]. We are furthermore grateful to Felix Kahlhoefer for his constructive and useful comments on the manuscript which allowed us to improve it.

**Funding information**  SA acknowledges support from the German Research Foundation (DFG) under grant No. AR 1321/1-1.

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
