# Peer review of "Benchmarking LHC searches for light 2HDM+a pseudoscalars"

_SciPost Physics, doi:SciPost Phys. 13, 007 (2022)_

## Round 1 · Referee Report · Anonymous (Referee 1) · 2022-4-16

The authors discuss an extension of the Standard Model (SM) usually referred to as 2HDM+a consisting of an extension of the scalar sector of the SM by an extra scalar doublet and an extra singlet (CP-odd). The model also includes a DM candidate in the form of a Dirac fermion, a singlet under the SM gauge group. The goal of the paper is to understand how specific processes constrain the model especially in the scenario where the pseudoscalar is very light. There were previous studies dealing with the same issue, like for instance 1802.02156, where one of the present authors participated. Two benchmark scenarios are used. I believe the paper is interesting although according to the scope of the paper it should be moved to SciPost Physics Core as it is not groundbreaking work. There are a few points that should be clarified before publication:

- The paper seems to very much built around the idea that eq. (4) makes a significant difference in the interpretation of the results of the searches. So my question is: had you included all constraints and all searches but not (4) would you get a sizeable difference in Figs. 1 and 2, left? Do the plots presented complete in any way what was presented in 1802.02156 by extending the mass region and by including new searches?

- Related to the previous comment, a short discussion about what happens to Figs. 1 and 2 left, if we move away from the benchmark points, is needed. I see that most of the light masses are excluded (except in regions where searches were not performed). Is this result robust against variation of the parameters?

- The authors state in the end of page 2 "For values of $|g_{haa}|$ that are not fine-tuned"; what does this sentence mean? And in the same paragraph can the authors please explain how the invisible Higgs decay width sets a lower bound of 100 GeV in the $a$ mass? Or did I misunderstand the sentence?

- When the authors start discussing benchmark I, they write "This corresponds to a parameter tuning of around 5 %". Can the authors explain the meaning of this sentence? Same question for benchmark 2.

I believe that the paper should be published in SciPost Physics Core after these issues are clarified.

---

## Round 1 · Referee Report · Anonymous (Referee 2) · 2022-5-5

Strengths

Overall the paper is well written and presents a clear case for the suggested benchmarks.

Weaknesses

The work is interesting however it is not clear what is new in and different from earlier work such as ( arXiv: 1802.02156 or 1711.02110)

Report

The authors analyze constraints on a Two Higgs doublet model with an addition pseudoscalar (2HDM+a) from existing LHC searches. Specifically they propose two possible benchamarks that can be used in experimental analyses. The work is interesting however it is not clear what is new in and different from earlier work such as ( arXiv: 1802.02156 or 1711.02110) . The work is not novel enough to be published in Scipost Physics, but possibly in Scipost Physics Core.

Requested changes

  1. The authors look at constraints from Higgs decays, how about the 4 lepton final states that would be relevant for larger masses of a (c.f. arXiv: 2107.00404)
  2. Given the large branching ratio to b-quarks, there should be some comments regarding the low and high mass dijet searches. (c.f arXiv: 1802.06149 (Fig 4) and arXiv: 1801.08769 )
  3. Minor: Typo on page 3 below equation 5 "...on $|g_{ahh}|$" assuming a light pseudoscalars a." should be "...on $|g_{haa}|$" assuming a light pseudoscalar a."
  4. Minor Suggestion: Orange and red are not the best colours to choose especially when there are overlapping regions - Could you possibly change or improve this presentation in the two Figures (1 and 2 right).

  • validity: good
  • significance: good
  • originality: low
  • clarity: good
  • formatting: good
  • grammar: excellent

Author:  Ulrich Haisch  on 2022-05-12  [id 2456]

(in reply to Report 2 on 2022-05-05)
Category:
answer to question

Please find attached a reply to the three referee reports.

Attachment:

reply.pdf

---

## Round 1 · Referee Report · Anonymous (Referee 3) · 2022-5-7

# REFEREE REPORT

| | |
|---|---|
| Authors: | Spyros Argyropoulos and Ulrich Haisch |
| Title: | Benchmarking LHC searches for light 2HDM+a pseudoscalars |
| Journal: | SciPost Physics |
| Preprint link: | scipost_202203_00007v1 |

I recommend publication of this paper in SciPost Physics after a minor revision of the manuscript following the points outlined below.

In general the analysis in Section 3 appears to be comprehensive while providing useful benchmarks for future LHC studies. I would have preferred a slightly more detailed presentation in Section 2 to make the paper more self-contained. The authors should at least provide a little more information on the relevant model parameters. For example, eq. (2) depends on a number of scalar potential parameters whose definitions are provided in a previously published paper. It would not have taken much space to present the relevant scalar potential prior to exhibiting eq. (2), so that the reader can immediately see the origin of the relevant parameters. The parameter $\theta$ is also introduces as a mixing angle, which again is related to the scalar potential parameters. To summarize, the relevant parameters (especially those used to define the benchmark scenarios) should be more explicitly defined in Section 2.

I also note that the authors denote the fermionic dark matter by $\chi$, although this symbol does not show up until the bottom of page 2. For clarity, this symbol should be introduced earlier. Finally, the statement at the top of page 3 on the lower limit of 100 GeV for $m_a$ is quite mysterious, as the authors associate this with the latest searches for invisible Higgs decay. However, for $a$ masses above half the Higgs boson mass (62.5 GeV), the invisible Higgs decay limits are irrelevant. Thus, it is hard to understand where the 100 GeV limit is coming from. (This point has also been echoed by one of the other referees.)

Finally, above eq. (3), the authors state that the total Higgs decay width predicted by the SM is 4.07 GeV, which is much smaller than the LHC sensitivity on $\Gamma_h$. The authors then impose $\Gamma_h < 1.1$ GeV. However, an indirect determination of the Higgs width at the LHC using off-shell Higgs production (admittedly with some caveats), concludes that the observed Higgs width is quite close to its Standard Model value. See arXiv:2202.06923 for further details. Presumably, this would lead to a significant reduction of the bound quoted in eq. (4). The authors should comment on this and indicate how the results of Section 3 would be affected by imposing the stricter bound.

Once these points are adequately addressed, I would be happy to support the publication of this work in SciPost Physics.

---

## Round 2 · Referee Report · Anonymous (Referee 1) · 2022-6-12

Report

The authors have answered all questions and the paper is ready for publication.

---

## Round 2 · Referee Report · Anonymous (Referee 3) · 2022-6-13

Report

In the revised manuscript, the authors have adequately addressed all the points raised in my initial referee report. Consequently, I am now ready to recommend the publication of this paper in SciPost Physics.

---

## Round 2 · Author Response

Please see the author reply id 2456

---

## Editorial Decision

published